# Molecular weight fractionation by confinement of polymer in one-dimensional pillar[5]arene channels

Tomoki Ogoshi [1,2,3], Ryuta Sueto[1], Masafumi Yagyu[1], Ryosuke Kojima[1], Takahiro Kakuta[1,2],
Tada-aki Yamagishi[1], Kazuki Doitomi[4], Anil Kumar Tummanapelli[5], Hajime Hirao[3,4], Yoko Sakata[1,2],
Shigehisa Akine[1,2] & Motohiro Mizuno[1]

Confinement of polymers in nano-spaces can induce unique molecular dynamics and properties. Here we show molecular weight fractionation by the confinement of single polymer chains of poly(ethylene oxide) (PEO) in the one-dimensional (1D) channels of crystalline pillar[5]arene. Pillar[5]arene crystals are activated by heating under reduced pressure. The activated crystals are immersed in melted PEO, causing the crystals to selectively take up PEO with high mass fraction. The high mass fractionation is caused by the greater number of attractive CH/π interactions between PEO C-H groups and the π-electron-rich 1D channel of the pillar[5]arene with increasing PEO chain length. The molecular motion of the confined PEO (PEO chain thickness of ~3.7 Å) in the 1D channel of pillar[5]arenes (diameter of ~4.7 Å) is highly restricted compared with that of neat PEO.

[1] Graduate School of Natural Science and Technology, Kanazawa University, Kakuma-machi, Kanazawa 920-1192, Japan. [2] WPI Nano Life Science Institute, Kanazawa University, Kakuma-machi, Kanazawa 920-1192, Japan. [3] Japan Science and Technology Agency (JST), Precursory Research for Embryonic Science and Technology (PRESTO), 4-1-8 Honcho, Kawaguchi, Saitama 332-0012, Japan. [4] Department of Chemistry, City University of Hong Kong, Tat Chee Avenue, Kowloon, Hong Kong, China. [5] Division of Chemistry and Biological Chemistry, School of Physical and Mathematical Sciences, Nanyang Technological University, 21 Nanyang Link, Singapore 637371, Singapore. Correspondence and requests for materials should be addressed to T.O. (email: ogoshi@se.kanazawa-u.ac.jp)

Confinement of polymers in nanometer scale spaces can increase understanding of the intrinsic properties of single polymer chains, control the molecular motion of polymer chains and provide new polymer properties. Zeolites, mesoporous silicas, organic crystal hosts, and porous coordination polymers/metal–organic frameworks have been used as nanoporous materials to confine polymers[1–8].

Pillar[n]arenes, which were introduced by our group in 2008[9], are pillar-shaped macrocyclic hosts that feature methylene bridge linkages at the para-positions of 2,5-dialkoxybenzene constituent units[10–17]. Their pillar-shaped structures provide high π-electron density inside their cavities, giving rise to excellent host–guest properties. Pillar[n]arenes can form host–guest complexes with not only cationic guests via charge-transfer interactions, but also neutral guests through multiple CH/π interactions[11,18]. However, almost all host–guest complexation experiments using pillar[n]arenes have been carried out in solution systems. Recently, we investigated crystal-state host–guest complexation using pillar[n]arenes[19–22]. We discovered that host–guest complexation could be achieved using pillar[n]arene crystals that were activated by drying solvates in the crystals by heating under reduced pressure. When the activated crystals were immersed in bulk guest solution, the angstrom-scale pillar[5]arene cavity rapidly took up guest molecules[22]. In this study, we explore the complexation of activated crystals of cyclic pentamer pillar[5]arene (**P5**) with a polymer, poly(ethylene oxide) (PEO). In the normal solution systems, no host–guest interactions of PEO with **P5** are observed. In contrast, immersing activated crystals of **P5** in melted PEO at 80 °C produces host–guest complex crystals denoted as **P5** ⊃ PEO complex. In the complex, PEO chains are included in the one-dimensional (1D) channel of the pillar[5]arenes in a crystalline state. It is interesting to note that these pillar[5]arene crystals form complexes selectively with high mass fraction of PEO when immersed in PEO with high polydispersity. This result demonstrates that the crystal-state complexation can be used to fractionate high mass fractions from polymer mixtures with a wide molecular weight distribution. The high mass fractionation using host–guest chemistry is an important achivement because high-molecular-weight polymers generally exhibit superior characteristics such as increased thermal stability, improved mechanical properties, and high crystallinity compared with low-molecular-weight polymers. The dynamic behavior of single polymer chains in the 1D channel is also investigated. The motion of PEO confined in the 1D channel of **P5** is highly restricted because the diameter of the 1D channel of **P5** (ca. 4.7 Å) fits to thickness of the PEO chain (ca. 3.7 Å).

## Results

**Encapsulation of PEO in the 1D channel of crystalline P5.** The host–guest complexation between **P5** (Fig. 1a) and PEO (Fig. 1b) in solution was investigated by [1]H NMR spectroscopy (Supplementary Fig. 2). When PEO with OH end groups ($M_n = 1000$, PEO1000-OH, where 1000 is molecular weight of PEO and OH is end groups of PEO) was mixed with **P5** in CDCl$_3$, neither peak shifts nor new peaks were observed, indicating that host–guest complexation hardly took place in CDCl$_3$. When n-alkanes were used as the guest, complexation occurred in CDCl$_3$[23,24]. Thus, the negatively charged oxygen atoms in the PEO chain inhibited the complexation in this case[25]. Next, we investigated the host–guest complexation of PEO with crystals of **P5**. First, crystals of **P5** were activated according to the reported method[19]. PEO melted when heated at 80 °C. The activated **P5** crystals were immersed in an excess amount of melted PEO at 80 °C (Fig. 1c; we determined a suitable PEO/**P5** feed ratio, Supplementary Fig. S4). Crystals of **P5** were insoluble in melted PEO, which

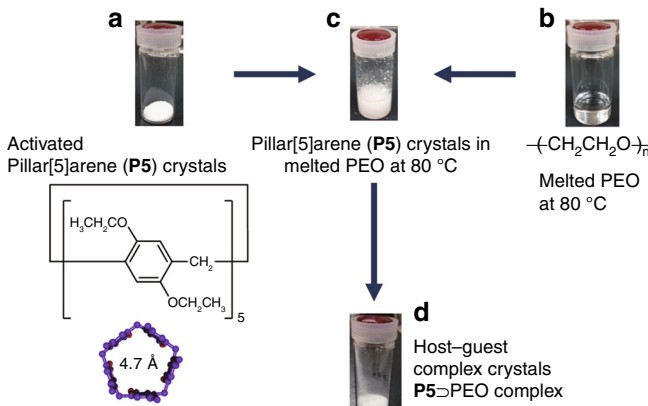

**Fig. 1** Polymer encapsulation using activated pillar[5]arene crystals. **a** Activated crystals of pillar[5]arene (**P5**). **b** Poly(ethylene oxide) (PEO) melted at 80 °C. **c**, **d** Host–guest complex crystals **P5** ⊃ PEO produced by immersing activated crystals of **P5** in melted PEO at 80 °C followed by washing the crystals with water to remove un-complexed PEO

means that the complexation proceeded in a heterogeneous system. After immersion in PEO, the crystals were isolated from the mixture by filtration and then washed with water to remove un-complexed PEO (Fig. 1d). The components of the host–guest complex crystals were dissolved in CDCl$_3$ (PEO and **P5** are all highly soluble in CDCl$_3$) and then investigated by [1]H NMR spectroscopy. Proton signals from PEO were clearly observed (Supplementary Fig. 3), despite the fact that the crystals were washed with water to remove un-complexed PEO, indicating that activated crystals of **P5** took up PEO during the immersion. As shown in Supplementary Fig. 2, **P5** hardly formed any host–guest complexes with PEO in solvent. Therefore, the crystalline-state complexation is a useful method to obtain host–guest complex crystals containing PEO. We monitored the time-dependent uptake of PEO in the crystalline-state complexation using [1]H NMR spectroscopy (Fig. 2a, Supplementary Fig. 3 and Supplementary Note 1).

Rapid uptake of PEO was observed for activated crystals of **P5**; it took ca. 10 min to reach an equilibrium state. In the equilibrium state, the PEO uptake amount ratio (PEO unit/host molar ratio) for **P5** crystals was ca. 4.0. The efficiency of the complexation of **P5** with PEO was also determined by organic vapor sorption experiments. Activated crystals of **P5** took up one n-hexane molecule per cavity upon exposing activated crystals of **P5** to n-hexane vapor[19,20]. However, **P5** ⊃ PEO complex crystals did not take up n-hexane vapor (Supplementary Fig. 5), indicating that all cavities in the **P5** crystals after PEO uptake are filled with PEO chains. The **P5** ⊃ PEO complex crystals were washed with water to ensure that there was no leakage of PEO from the pillar[5]arene cavities.

The complexation of PEO with **P5** in crystalline state was confirmed directly by magic angle spinning 2D heterocorrelated NMR study (Fig. 2b). The cross peak observed can be assigned to inter-molecular host–guest correlations of the CH$_3$ proton groups of **P5** and the carbon atoms of PEO, indicating the inclusion of PEO chain in the cavity of **P5**. From the previous reports[26,27], carbon signals of PEO in non-crystalline and crystalline phases were observed at 72.0 and 71.2 ppm, respectively. The chemical shift of neat PEO was observed at 72.0 ppm (Fig. 2c, top), indicating formation of crystalline phase. In contrast, the chemical shift observed in **P5** ⊃ PEO complex crystals (Fig. 2c, bottom) was ca. 70 ppm, which was remarkably lower than 71.2 ppm and 72.0 ppm assigned to PEO carbon in non-crystalline

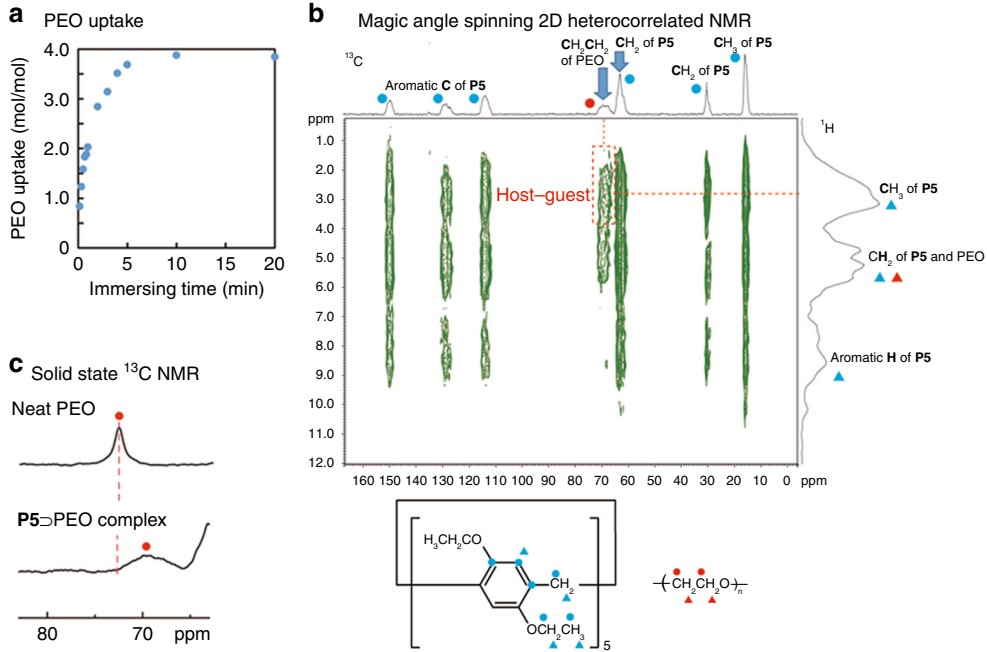

**Fig. 2** Encapsulation of polymer in 1D pillar[5]arene channels. **a** Uptake ratio of poly(ethylene oxide) (PEO1000-OH) by activated crystals of pillar[5]arene **P5** (blue circles). **b** Magic angle spinning 2D heterocorrelated NMR spectrum of **P5** ⊃ PEO complex with PMLG homonuclear decoupling (ct = 1 ms). **c** Solid-state $^{13}$C NMR spectra of neat **PEO** at 32 °C (top) and the **P5** ⊃ PEO complex at 30 °C (bottom)

and crystalline phases, respectively. Therefore, the upper chemical shift of the **P5** ⊃ PEO complex crystals would result from aromatic shielding of PEO upon incorporation into the cavities of **P5**.

To investigate the assembled structures of **P5** ⊃ PEO complex, powder X-ray diffraction patterns (PXRD) were measured. The PXRD pattern of **P5** ⊃ PEO complex crystals was quite different from that of activated **P5** crystals (Supplementary Fig. 7), indicating that a crystal transformation underwent by the PEO uptake. PXRD patterns of the host–guest complex crystals using **P5** can be classified as two major classes, i.e., 1D channel and herringbone structures[22]. As a rule, when the length of the *n*-alkanes was increased, **P5** ⊃ *n*-alkane complexes changed from a herringbone to a 1D channel structure to cover the protruding part of the *n*-alkane molecule. **P5** ⊃ *n*-hexane complex forms herringbone structure, but **P5** ⊃ *n*-octane complex forms 1D channel structure. The PXRD pattern of **P5** ⊃ PEO complex in low angles was similar to that of **P5** ⊃ *n*-octane complex, but not similar to that of **P5** ⊃ *n*-hexane complex. These data indicates that **P5** molecules in the crystals after the PEO uptake also assembled to form 1D channels containing PEO chains. PEO is long linear chain guest, thus formation of the 1D channels should be reasonable.

**Molecular weight fractionation by activated P5 crystals**. Activated crystals of **P5** are able to take up PEO when they are immersed in melted PEO. Next, we turned our attention to the dependence of PEO uptake efficiency on the molecular weight of PEO. PEO uptake time was monitored by $^{1}$H NMR spectroscopy. As the molecular weight of PEO increased, complexation took a longer time to reach the equilibrium state (Supplementary Fig. 8). To form a complex, an end group of PEO needs to thread into a **P5** crystal. The number of end groups of PEO decreases as the molecular weight of PEO increases. Therefore, the complexation took a longer time to form as the molecular weight of PEO increased.

We investigated the uptake of PEO with different molecular weights by activated crystals of **P5**. **P5** ⊃ PEO complex crystals containing PEO10000-OH were prepared by immersing activated crystals **P5** in melted PEO1000-OH. Subsequently, the **P5** crystals complexed with PEO1000-OH were immersed in melted PEO1000-OH. The molecular weight of PEO encapsulated in the host–guest complex crystals was evaluated by liquid chromatography. Peaks from PEO10000-OH were observed, whereas those from PEO1000-OH were not (Supplementary Fig. 9 and Supplementary Note 3), indicating that the guest PEO1000-OH was replaced with PEO10000-OH. In contrast, a similar guest replacement event did not occur when **P5** ⊃ PEO complex crystals containing PEO10000-OH were immersed in melted PEO1000-OH (Supplementary Fig. 10 and Supplementary Note 4). These results indicate that activated **P5** crystals interact more strongly with PEO of higher mass fraction than with PEO of lower mass fraction. Next, activated **P5** crystals were immersed in an equal-weight mixture of PEO with different molecular weights (Fig. 3a, top, PEO1000-OH, PEO4000-OH, PEO6000-OH, and PEO10000-OH). The host–guest complex crystals formed after the immersion showed peaks from PEO with high mass fraction, mainly PEO6000-OH and PEO10000-OH (Fig. 3a, bottom), indicating that activated **P5** crystals selectively took up PEO with high mass fraction from the PEO mixture with a wide molecular weight dispersion. The selectivity was independent of the immersion time (Supplementary Fig. 11). When activated crystals of **P5** were immersed in another equal-weight mixture of PEO (Fig. 3b, top, PEO1000-OH, PEO4000-OH, and PEO6000-OH), **P5** crystals selectively took up PEO with a high mass fraction, mainly PEO6000-OH (Fig. 3b, bottom). The high mass fractionation was also confirmed by $^{1}$H NMR (Supplementary Fig. 12 and Supplementary Note 5). These results demonstrate that activated crystals of **P5** are able to sort PEO with high mass fraction from PEO mixtures with high polydispersity.

To better understand the molecular weight fractionation, we calculated binding energies of various model complexes using

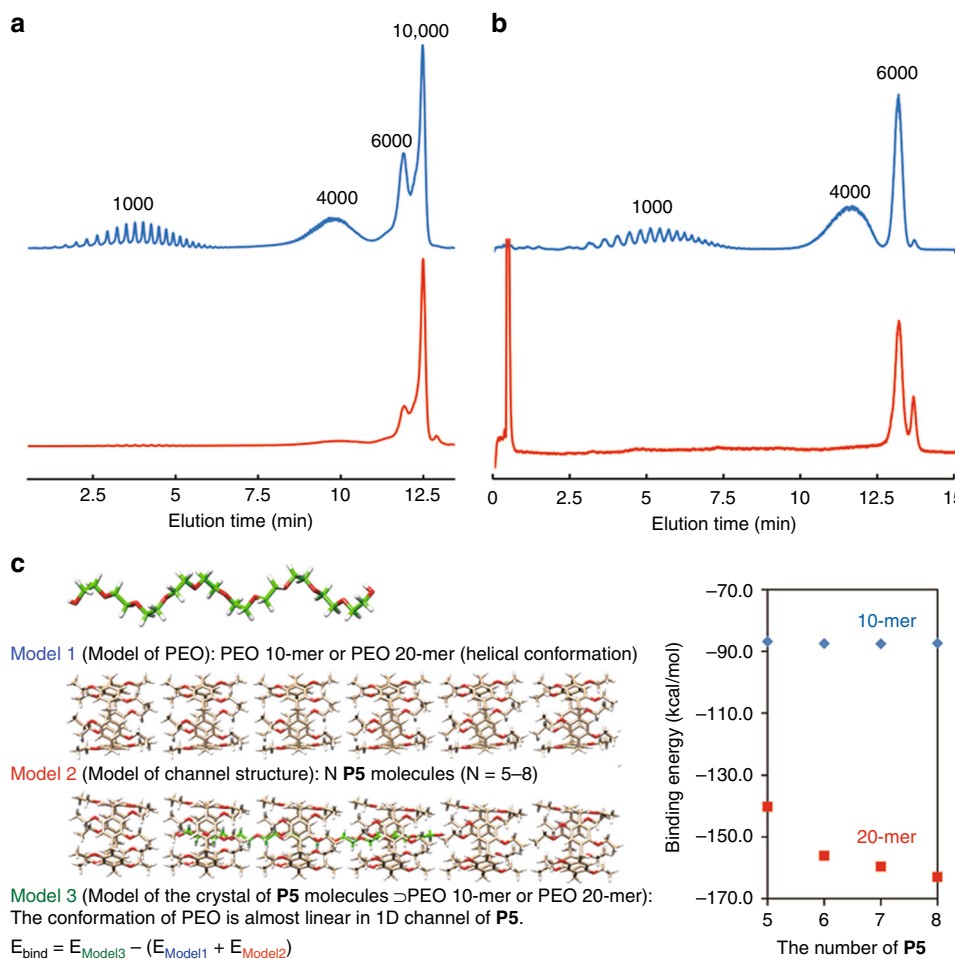

**Fig. 3** High mass fractionation by 1D pillar[5]arene channels. Liquid chromatography traces of an equal-weight mixture of PEO (blue lines, **a** PEO1000-OH, PEO4000-OH, PEO6000-OH, and PEO10000-OH) and **b** PEO1000-OH, PEO4000-OH, and PEO6000-OH) and **P5** crystals after immersion in the mixture (red lines). **c** Calculated binding energies for PEO 10-mer and PEO 20-mer using model 1–3

molecular mechanics (MM) simulations (Fig. 3c). The binding energies between a PEO 10-mer and model 1D channel constructed from pillar[5]arene molecules were almost constant (blue diamonds, −86.8 to −87.6 kcal/mol) and did not increase even when the number of pillar[5]arene molecules in the 1D channel was increased from five to eight. This is because the PEO 10-mer is already fully covered by the 1D channel constructed from five pillar[5]arene molecules (Supplementary Fig. 13), so a further increase of the number of pillar[5]arenes molecules did not provide additional stabilization. However, as the number of pillar[5]arene molecules was increased from five to eight, the binding energy between a PEO 20-mer and the 1D channel increased accordingly from −140.3 to −163.0 kcal/mol (red squares). The PEO 20-mer was longer than the 1D channel of five pillar[5]arene molecules, so the PEO 20-mer was not completely covered by pillar[5]arenes (Supplementary Fig. 13). In contrast, the PEO 20-mer was fully covered by the 1D channels constructed from eight pillar[5]arene molecules (Supplementary Figs 14–16), thus resulting in a larger stabilization effect. This explains why the binding energy of the PEO 20-mer changed noticeably with the length of the 1D channel. Multiple attractive van der Waals interactions exist between a PEO chain and a 1D channel. Therefore, effective coverage of PEO by a channel results in a high binding energy. In addition, the PEO 20-mer can form a larger number of interactions with a pillar[5]arene channel than

the PEO 10-mer, even when the channel length is the same. Consequently, the binding energy of the PEO 20-mer is always larger than that of the PEO 10-mer; that is, the longer PEO 20-mer binds more strongly to the 1D channel than the PEO 10-mer. The calculation results are thus consistent with the observed high mass fractionation.

**Effect of end groups on PEO uptake by activated P5 crystals.** We also investigated the effect of the end groups of PEO on its PEO uptake by activated crystals of **P5**. PEO with methoxy (OMe), amine ($NH_2$), tosyl, and carboxylic acid (COOH) end groups were used. The rapid complexation occurred in PEO with OMe ends (Fig. 4, 3 min to reach equilibrium), which was faster than that in PEO with OH end groups (10 min) and $NH_2$ end groups (20 min). In the cases of the OH and $NH_2$ end groups, OH/O and NH/O hydrogen-bonding interactions between the OH or $NH_2$ end groups of PEO and O atoms of **P5** may slow the threading of **P5** on the polymeric chains. In the case of PEO with COOH end groups, the amount of PEO taken up was small (guest/host ratio = ca. 1), and the PEO uptake did not reach the saturated state even after 3 h. The cavities of pillar[5]arenes are electron-rich; therefore, electronic repulsion between the electron-rich pillar[5]arene cavities and electron-rich carbonyl groups of the COOH end groups would hinder the complexation.

The same behavior was reported for the host–guest complexes formed between a viologen guest containing COOH end groups and a pillar[5]arene[28]. When we used PEO with tosyl end groups, the PEO uptake was slow compared with that of PEO with OH end groups. Tosyl end groups are bulky compared with OH ones, so steric hindrance would slow complexation in this case.

**Dynamics of PEO confined in the 1D channel of crystalline** P5. We envisioned that the encapsulation may also influence the thermal transition behavior and dynamics of PEO. To evaluate the thermal transition behavior of PEO, differential scanning calorimetry (DSC) measurements were performed (Fig. 5a).

The DSC scans for neat PEO and activated **P5** crystals yielded endothermic peaks at 40 and 156 °C, respectively, corresponding to the melting temperatures of these materials. In contrast, the **P5** ⊃ PEO complex crystals did not show these endothermic peaks; instead, a new endothermic peak appeared at 179 °C. The data indicates that PEO confined in the 1D channel of pillar[5] arenes exhibits different thermal transition behavior compared with that of neat PEO. The dynamics of PEO confined in the 1D channel was evaluated by solid-state cross polarization-magic angle spinning (CP-MAS) $^{13}$C NMR spectroscopy (Fig. 5b). Peaks corresponding to PEO chains accommodated in the channels were detected at around 70 ppm (Fig. 2c). For neat PEO, the half-width at half-maximum (HWHM) values of the PEO peak were around 50–100 Hz (filled circles). In stark contrast, the HWHM values for PEO encapsulated in the 1D channel of **P5** were ca. 300 Hz, and remained almost constant at different temperatures (open circles). This indicates that the mobility of a PEO chain in the 1D channel of **P5** was much lower than that of a PEO chain in the neat environment. We also measured variable temperature $^{13}$C spin-lattice relaxation time ($T_1$) to investigate the mobility of PEO (Fig. 5c). For the $T_1$ measurements of the **P5** ⊃ PEO

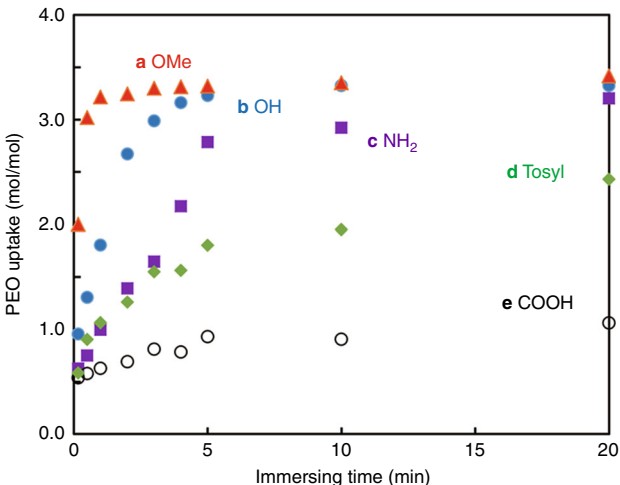

**Fig. 4** Effect of end groups on PEO uptake. Uptake ratio of PEO with **a**, OMe (PEO1000-OMe), **b** OH (PEO1000-OH), **c** NH$_2$ (PEO1000-NH$_2$), **d** tosyl (PEO1000-Ts), and **e** COOH (PEO1000-COOH) end groups by activated crystals of **P5**

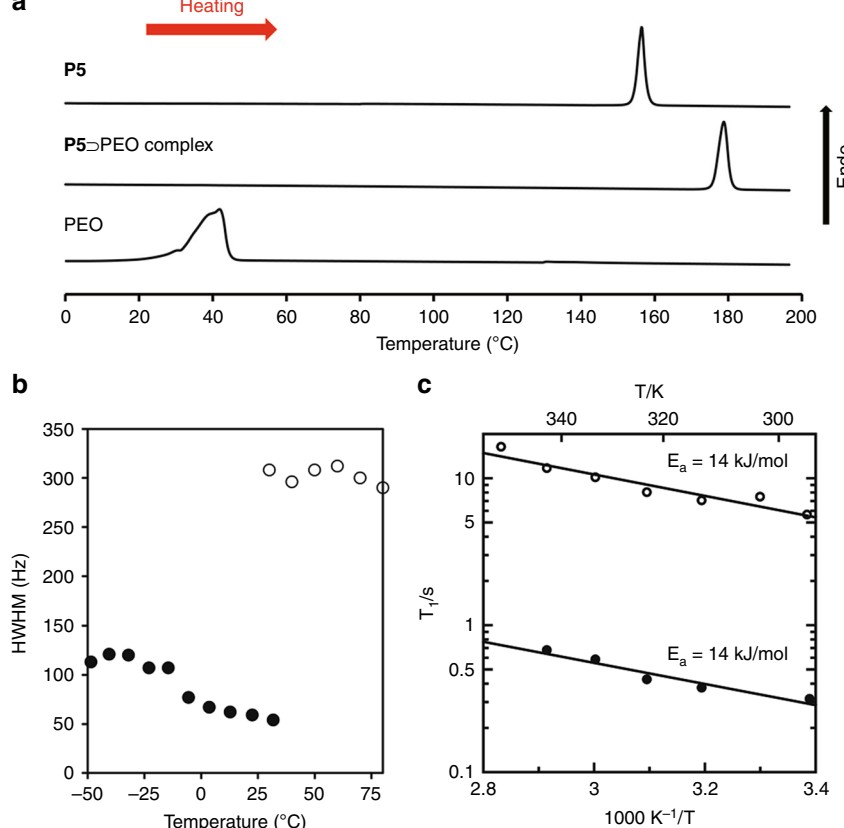

**Fig. 5** Thermal transitions and molecular dynamics of PEO confined in 1D pillar[5]arene channels. **a** DSC heating curves of **P5**, **P5** ⊃ PEO, and PEO. PEO1000-OH was used. **b** Plots of the half-width at half-maximum (HWHM) values of the signal from complexed and un-complexed PEO (Fig. 2b) at different temperatures (open circles = **P5** ⊃ PEO complex and filled circles = neat PEO). **c** Arrhenius plot of $^{13}$C $T_1$ as a function of temperature (open circles = **P5** ⊃ PEO complex and filled circles = neat PEO)

complex crystals, we used CP-MAS method. However, we could not use CP-MAS method for the $T_1$ measurements of neat PEO. Therefore, we performed DD-MAS method for the $T_1$ measurements of neat PEO, which indicated that mobility of neat PEO was remarkably higher than PEO in the 1D channels of **P5**. In the measurement temperatures, $T_1$ values of neat PEO were shorter than that of the **P5** ⊃ PEO complex crystals, indicating that neat PEO has higher mobility than PEO in the 1D channels of **P5**. The data also suggests that the mobility of a PEO chain in the 1D channel of **P5** was much lower than that of a PEO chain in the neat environment. We performed molecular dynamics (MD) simulations of a model PEO molecule in an isolated state and in a 1D channel of pillar[5]arenes. The torsion angles of PEO in the MD trajectories were also analysed. The MD studies showed that PEO in **P5** favors the all-anti conformation (Supplementary Fig. 19). The simulation results therefore suggest that PEO is considerably restrained within the pillar[5]arene channel, which is consistent with the dynamics of PEO measured by solid-state $^{13}$C NMR.

## Discussion

We demonstrated the unique complexation behavior of the linear polymer PEO with activated pillar[5]arene crystals. In view of the fact that the complexation did not occur in a common host–guest complexation system using a solvent, the crystalline-state complexation presented in this study offers a useful strategy that allows complexation between PEO and pillar[5]arenes, although complexation of specific cationic polymers with pillar[n]arenes in solution has been reported[29,30]. Various types of PEO are commercially available, and PEO has high functionality and good solubility in various solvents; thus we believe that the poly-pseudorotaxanes consisting of pillar[5]arenes and PEO should be useful for material applications.

The activated pillar[5]arene crystals selectively took up PEO with high mass fraction from mixtures of PEO with high polydispersity. Our MM calculation indicated that the formation of multiple CH/π interactions between the PEO chain and pillar[5]arene 1D channel increases the binding energy of the complex, which contributes to the selectivity for high mass PEO sorting. The end groups of PEO influenced its uptake. Complexation was slower for PEO with OH and NH$_2$ end groups compared with that of PEO with OMe end groups. The uptake of PEO with COOH and tosyl end groups was slow compared with that of PEO with OH end groups.

Thermal transitions of PEO in the 1D channels of **P5** occurred at higher temperature compared with that of neat PEO. The motion of PEO confined in the 1D channel of **P5** was extremely limited compared with neat PEO because PEO chain fitted snugly into the 1D channel of **P5**. Understanding the thermal transition and dynamic behavior of a single polymer chain encapsulated in the channel of a pillar[5]arene should have considerable implications for their potential application as single polymer transporters, lubricants, and connecters in molecular-based devices. In this respect, we believe that our results provide new insights into the polymer dynamics within a 1D channel.

## Methods

**Materials**. All solvents and reagents were used as supplied. **P5** was synthesized according to the previous paper[31]. Supplementary Table S1 shows product information of poly(ethylene oxide) (PEO) used in this study. PEO with COOH ends (PEO-1000-COOH) was prepared from PEO1000-OH according to the previous paper[32].

**Solution NMR**. Solution $^1$H NMR spectra were recorded at 500 MHz with a JEOL-ECA500 spectrometer.

**Solid NMR**. Solid-state $^{13}$C NMR spectrum was measured using a JEOL ECA-300 spectrometer operating at 74.175 MHz. High-resolution solid-state NMR spectrum was obtained using magic-angle spinning (MAS) and high-power $^1$H dipole decoupling (DD). Cross-polarization (CP) was used for signal enhancement. The sample was packed into a 4 mm diameter zirconia rotor. The total suppression of sidebands (TOSS) sequence was used to suppress spinning sidebands[33]. The MAS rate was set to 5 kHz. $^{13}$C chemical shifts were expressed as values relative to tetramethylsilane (TMS) using the 29.50 ppm line of adamantane as an external reference. The Torchia pulse sequence was used for $^{13}$C spin-lattice relaxation time ($T_1$) measurement of **P5** ⊃ PEO complex[34]. The inversion recovery method based on DD-MAS was used for $^{13}$C $T_1$ measurement of neat PEO. In the $^1$H–$^{13}$C heterocorrelated 2D NMR measurement, the $^1$H signals were detected using Phase Modulated Lee–Goldburg (PMLG) homonuclear decoupling[35].

**Powder X-ray diffraction**. Laboratory powder X-ray diffraction (PXRD) measurements were performed on a Rigaku Smart Lab high-resolution diffractometer.

**Organic vapor sorption experiments**. Organic vapor sorption isotherms were obtained by a BELSORP-max (BEL Japan Inc., Osaka, Japan) sorption analyzer.

**Liquid chromatography**. Liquid Chromatography (GC) analyses were performed on a Thermo Fisher Scientific chromatograph (UltiMate3000 Thermo Fisher Scientific column: Accucore C18) with a Corona VeoRS detection. Mixture of water and acetonitrile were used as eluents. The gradient program of eluents for molecular weight distribution of PEO is shown in Supplementary Fig. 1.

**Activation of P5 crystals**. Crystals of **P5** were activated according to the previous report[19]. **P5** was dissolved in acetone, and then the evaporation of acetone afforded crystals **P5**. Drying crystals **P5** at 80 °C under reduced pressure for 24 h was sufficient to de-solvate all the acetone molecules and afforded activated crystals of **P5**. Removing acetone was confirmed by $^1$H NMR measurement.

**Complexation of PEO with activated P5 crystals**. Activated crystals of **P5** (64 mg) were immersed in a sealed 5 mL vial containing melted PEO (500 mg) at 80 °C. The feed ratio [PEO/**P5** = 500/64 (wt/wt)] was determined by evaluating the feed ratio effect on the PEO uptake time-dependency and amount at equilibrium state (Supplementary Fig. 4 and Supplementary Note 2). The heterogeneous mixture was heated at 80 °C. To remove un-complexed PEO, the crystals were washed with water. The amount of PEO taken up by activated crystals of **P5** was monitored by completely dissolving the crystals in CDCl$_3$ and measuring the ratio of proton signals from ethylene moieties of PEO to that from phenyl groups of **P5** by $^1$H NMR spectroscopy (Fig. 2a and Supplementary Fig. 3). **P5** ⊃ PEO host–guest complex crystals for vapor sorption experiments (Supplementary Fig. 5), PXRD (Supplementary Fig. 7), DSC, and solid state NMR (Fig. 5) were prepared by immersing activated crystals of **P5** in melted PEO1000-OH for 12 h.

**Competitive PEO uptake with different molecular weights**. 64 mg of the host–guest complex crystals of **P5** containing PEO1000-OH, which was prepared by immersing activated crystals of **P5** in melted PEO1000-OH at 80 °C for 12 h, was immersed in 500 mg of melted PEO10000-OH at 80 °C for 12 h. Molecular weights of PEO in the crystals after the immersion was investigated by liquid chromatography (Supplementary Fig. 9). 64 mg of the host–guest complex crystals of **P5** containing PEO10000-OH, which was prepared by immersing activated crystals of **P5** in melted PEO10000-OH at 80 °C for 12 h, was immersed in 500 mg of melted PEO1000-OH at 80 °C for 12 h. Molecular weights of PEO after the immersion was investigated by liquid chromatography (Supplementary Fig. 10).

**Molecular weight fractionation using activated P5 crystals**. Activated **P5** crystals (50 mg) were immersed in a sealed 5 mL vial containing an equal-weight mixture of melted PEO with different molecular weights (160 mg × 4 in Fig. 3a, and 160 mg × 3 in Fig. 3b) at 80 °C for 12 h. To remove uncomplexed PEO, the crystals were washed with water. The molecular weight of PEO molecules encapsulated in the host–guest complex crystals was evaluated by liquid chromatography (Fig. 3a, b). The selectivity was independent of the immersing time (Supplementary Fig. 11).

**Molecular mechanics (MM) calculations**. We used molecular mechanics (MM) to calculate binding energies between PEO (PEO 10-mer or 20-mer) and **P5**. For this purpose, the Forcite program implemented in Materials Studio 8.0 was used[36], and molecules were described by use of the COMPASS II (version 1.2) force field[37]. Models 1 (PEO 10-mer and 20-mer) were prepared from the crystal structure of PEO in the bulk state[38], and PEO is a helical structure in the bulk state. Models 2 for **P5** were built from the X-ray structure of **P5** ⊃ *n*-octane complex. In Models 3, PEO is in the cavity of **P5**. Binding energies were calculated using the following equation:

$$E(bind) = E(Model3) - \{E(Model1) + E(Model2)\}$$

where the E values are energies of geometry-optimized molecules.

**Conformations of PEO in P5**. To find a stable conformation of PEO in pillar[5]arene, we performed additional MM calculations. Here, we evaluated the binding energy between 3 pillar[5]arenes and PEO 10-mer in the linear or helical geometry. The linear geometry has all-anti conformations at heavy atoms, while the helical conformation contains gauche conformations. In the case of **P5**, the binding energy was calculated as −87.8 kcal/mol in the linear geometry, while the binding energy was −82.8 kcal/mol in the helical geometry. These results indicate that PEO favors the linear conformation within **P5**.

**Molecular dynamics (MD) simulations**. For molecular dynamics (MD) simulations, we used a short PEO model (Supplementary Fig. 17). Periodic models of free PEO and **P5** ⊃ PEO complex were also built (Supplementary Fig. 18). The free PEO model contained 16 PEO molecules in a cubic cell with a size of $50 \times 50 \times 50$ Å$^3$. The supercell models of pillar[5]arene complexes were built from X-ray structures. All simulations were performed with the Forcite module of Materials Studio 8.0[37]. The COMPASS II (version 1.2) force field[37,38] was used with the force-field-assigned atomic charges.

Prior to MD simulations, we performed energy minimization calculations for 5000 cycles with the Smart algorithm. Subsequently, we performed MD simulations for 10 ns in constant NPT conditions. MD trajectories were analyzed to determine the distributions of different torsion angles (Supplementary Fig. 19).

## Data availability

The authors declare that all data supporting the findings of this study are available within the Article and its Supplementary Information. The raw data generated in this study are available from the corresponding author upon reasonable request.

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

## Acknowledgements

This work was partially supported by Grant-in-Aid for Scientific Research on Innovative Areas: π-System Figuration (JP15H00990 and JP17H05148) and Soft Crystals (JP18H04510 for T.O. and JP18H04511 for Y.S.), JST PRESTO (JPMJPR1313 for T.O. and JPMJPR141B for H.H.), JST CREST (JPMJCR18R3) and Kanazawa University CHOZEN Project. H.H. acknowledges financial support from City University of Hong Kong (7200534 and 9610369). NanoLSI is supported by the World Premier International Research Centre (WPI) Initiative, Japan.

## Author contributions

T.O. conceived the project and designed the experiments. R.S. M.Y., R.K., T.K. and T.Y. investigated crystalline state complexation with PEO; K.D., A.K.T. and H.H. performed M.M. and M.D. calculations; Y.S. and S.A. measured X-ray single-crystal structures; and M.M. performed the solid-state NMR measurements. All authors analysed and discussed the results, and co-wrote the paper.

## Additional information

**Competing interests:** The authors declare no competing interests.

