## [Peer Review File · Nature Communications]

Reviewers' comments:

Reviewer #1 (Remarks to the Author):

The study proposed Ogoshi et al. investigate the use of 'activated' crystals of per-ethylated pillar[5]arene and pillar[6]arene to selectively take up higher molecular chains from a polydisperse sample of PEO. The synthesis of the 'activated' crystals (i.e. the P5/P6 cavities are empty) had previously been reported by the same group and use for the intake of small alkanes. Here they investigate the capture of much longer chains and they explain the selectivity of intake by the greater number of C-H/ π interactions formed by long chains over shorter ones. This is a very interesting study that presents a new approach to polymer separation and, maybe more importantly, an insight on the behaviour of polymer chains in confined spaces.

Overall the paper is well presented and the claims are, for most part, well supported by the experiments (although the authors could be a bit more cautious with some of the claims, see below). This paper should be a great interest for the broad readership of Nature Communications.

Some comments:

Title: 'High-molecular-weight polymer sorting' can be misleading since the experiments have been performed on rather short PEO. I suspect the intention was to emphasize the fact that the high-mass fraction of a polymer mixture is taken by the P5/6 crystal but I'm not quite sure the title actually conveys this idea.

Line 74: 'PEO confined in the 1D channel of P6 had the same mobility as that of neat PEO' I would be a bit more cautious here. It appears from the data that the PEO has a local mobility comparable to neat PEO but it is still forced in an extended conformation (unlike neat PEO).

Figure 2: the full ^{13}C NMR spectra should be provided in the SI.

Figure 2: Caption and axis labels are a bit small.

Line 133: 'this binding mode is not possible in the P6 channel.' It's probably more appropriate to say that it's less favored.

Line 143-146 (+SI): The PXRD pattern of P5/PEO looks quite different from the P5/octane

Line 65: fractinate should be fractionate

Line 81: looking at table S1 it seems it should be $M_n = 1000$ rather than M_w . This should be checked across the manuscript and the SI as well.

Manuscript and SI should be thoroughly checked for spelling/typo errors.

Reviewer #2 (Remarks to the Author):

This paper deals with the formation of crystalline inclusion compounds of PEO. Confinement of PEO chains in crystalline inclusion compounds is not new and can be encompassed within the studies of polymer chains in 1D nanochannels. These studies originated in the eighties and nineties. In any case, the paper needs a profound revision and several additional data, according to the

following comments and suggestions:

TITLE : ' high mol-weights ' . actually, they cannot be considered high molecular weights, but oligomers. Please, erase the concept.

CONFORMATIONS UNDER CONFINEMENT

C13 NMR is the most powerful tool to detect conformations, dynamics and proof of inclusion .

Therefore, it should be enforced considerably with additional experiments. In fact, the demonstration and discussion about conformations in channels is not satisfactory.

The authors say: ' Both n-alkanes and PEO are linear, unbranched molecules. Thus, these data indicate that PEO prefer to adopt all-anti-like conformations in the PS channel.' THIS deductive procedure is rather debatable and the discussion too hermetic.

Instead, PEO chemical shifts inside the channels must be compared to PEO in the crystalline phase and amorphous phase (which differ considerably). The discussion should be conducted independently of n-hydrocarbons.

The authors state (line : 136): 'An up-field shift of the carbon signal from PEO was observed for the PSDPEO (Fig. 2b, middle) and P6DPEO complex crystals (Fig.

2b, bottom), which was caused by aromatic shielding of PEO upon incorporation into the complex parts in the crystals formed an assembled structure that was the same as that observed in the X-ray crystal structure of the P61) n-decane complex; i.e., 1D

channels (Fig. 2d). Likewise, P6 molecules also formed a 1D channel containing PEO.'

The analogy with alkanes is not stringent enough. PEO crystallizes in a gauche-containing conformation and its chemical shift in the crystalline state is up-field and not down-field. It must be disentangle clearly the effects of surrounding walls from the conformational arrangement of the macromolecular chains. Instead, n-alkanes crystallize in the all-trans conformation.. Additionally, PEO is not fully crystalline, but is necessarily partly amorphous. Which ones are taken as chemical shift for the stretched and the gauche-containing conformations. The MAS spectra, run with CP and without CP at various delay times between pulses, should be performed on the included and confined PEO.

Thus, the discussion of the literature and the papers dealing with the 13C NMR chemical shifts and the PEO confinement is to be revised and re-written, by providing more support to the conclusions. The comparison between alkanes of different chain-lengths is confusing and not relevant to the subject of the paper: therefore, the comments on the conformational arrangements of hydrocarbons should be erased.

MOLECULAR WEIGHTS

Molecular weights should be detectable by the analysis of the terminal groups. Solid state NMR should provide evidences between in and out PEO.

It is not clear whether conformations are affected by the molecular size and functional end groups.

MOBILITY

Line 252: 'half-width at half-maximum (HWHM) values of the peaks for PEO were around 100 Hz and nearly constant at different temperatures (blue squares). The HWHM values for neat PEO (green triangles) and PEO encapsulated in the 1D channel of P6 were almost the same. In stark contrast, the HWHM values for PEO encapsulated in the 1D channel of PS were ca. 300 Hz, and remained almost constant at different temperatures (red circles).

This indicates that the mobility of a PEO chain in the 1D channel of PS was much lower than that of a PEO chain in the 1D channel of P6 and in the neat environment.'

On the only basis of the line width is hard to establish the motional behavior: in other words interpreted as diverse T2 (being the reciprocal of the LW). This is a very weak evidence, because the line-width may depend on a few other phenomena differing from mobility.

Mobility is indicated as reduced in the channels . On the contrary, increased mobility of confined macromolecules occurs frequently in compatible channels. The literature on this topic is not discussed.

INCLUSION OF PEO IN LARGER CHANNELS

Additionally, evidence for the formation of IC in P6 with larger channels is weak.

In conclusion, I recommend that the authors fulfill the above requests.

According to two reviewer's comments, we have revised our manuscript. We would like to explain these corrections point-by-point toward each reviewer as follows:

Reviewer 1

We appreciate the recommendation of the reviewer to publish this paper. According to the comments of the reviewer, we revised our manuscript as follows:

Comment 1: Title: 'High-molecular-weight polymer sorting' can be misleading since the experiments have been performed on rather short PEO. I suspect the intention was to emphasize the fact that the high-mass fraction of a polymer mixture is taken by the P5/6 crystal but I'm not quite sure the title actually conveys this idea.

Answer: According to the reviewer's comment, we changed the title as follows:

High-mass fractionation by confinement of polymer in one-dimensional pillar[5]arene channels

Comment. 2 Line 74: 'PEO confined in the 1D channel of P6 had the same mobility as that of neat PEO' I would be a bit more cautious here. It appears from the data that the PEO has a local mobility comparable to neat PEO but it is still forced in an extended conformation (unlike neat PEO).

Answer:

[redacted]

[redacted]

More experiments need to clarify the complicated crystal structure effect on the uptake behaviour of PEO. However, about the PEO uptake using activated **P5** crystals, the uptake amount of PEO in activated **P5** crystals was independent of the experimental condition, and showed good reproducibility. Thus, we would like to focus on the complexation of PEO by activated crystals of **P5**, and remove the discussion about the complexation of PEO by activated crystals of **P6** in this communication. The removing the data using activated crystals of **P6** do not affect the main story of the paper. From the view point of the reproducibility, we believe that removing the experiments using **P6** should be true choice in the current stage. We would like to report the details next paper.

Comment. 3 Figure 2: the full ^{13}C NMR spectra should be provided in the SI.

Answer: According to the reviewer's comment, we provided the full ^{13}C NMR spectra in SI.

Comment. 4 Figure 2: Caption and axis labels are a bit small.

Answer: According to the reviewer's comment, we increased the size of caption and axis. According to the comment's reviewer 2, we removed the crystal structure of host-guest complex between *n*-alkane and **P5**, and added magic angle spinning 2D heterocorrelated NMR study.

Comment. 5 Line 133: 'this binding mode is not possible in the P6 channel.' It's probably more appropriate to say that it's less favored.

Answer: [redacted]

Comment. 6 Line 143-146 (+SI): The PXRD pattern of P5/PEO looks quite different from the P5/octane

Answer: From the previous investigation (Ogoshi et al. Chem Commun. 2017, 53, 8577), the PXRD patterns of the host-guest complex crystals using **P5** can be classified as two major classes, *i.e.*, 1D channel and herring bone structures. The PXRD pattern of **P5**⊃PEO complex is similar to that of 1D channel structures, but not similar to that of herringbone structures. We mentioned the point in the revised manuscript as follows:

PXRD patterns of the host-guest complex crystals using **P5** can be classified as two major classes, *i.e.*, 1D channel and herringbone structures.²¹ As a rule, when the length of the *n*-alkanes was increased, **P5**⊃*n*-alkane complexes changed from a herringbone to a 1D channels structure to cover the protruding part of the *n*-alkane molecule from the cavity of **P5**. **P5**⊃*n*-hexane complex forms herringbone structure, but **P5**⊃*n*-octane complex forms 1D channel structure. The PXRD pattern of **P5**⊃PEO complex is similar to that of **P5**⊃*n*-octane complex, but not similar to that of **P5**⊃*n*-hexane complex. These data indicates that **P5** molecules in the crystals after PEO uptake also assembled to form 1D channels containing PEO chains. PEO is linear long chain guest, thus formation of the 1D channels should be reasonable.

Comment. 7 Line 65: fractinate should be fractionate

Answer: The typo is corrected in the revised manuscript.

Comment. 8 Line 81: looking at table S1 it seems it should be Mn = 1000 rather than Mw. This should be checked across the manuscript and the SI as well.

Answer: We checked the description across the manuscript and the SI as well.

Comment. 9 Manuscript and SI should be thoroughly checked for spelling/typo errors.

Answer: We carefully checked the spelling/typo errors.

Reviewer 2

We appreciate the comments of the reviewer to improve this paper. According to the comments of the reviewer, we revised our manuscript as follows:

Comment 1: Confinement of PEO chains in crystalline inclusion compounds is not new and can be encompassed within the studies of polymer chains in 1D nanochannels. These studies originated in the eighties and nineties.

Answer 1: As the reviewer mentioned, confinement of PEO chain in inclusion compounds has been reported, but there are no report using pillar[n]arene crystals. In this paper, we would like to focus on the high mass fractionation by confinement of polymer by activated crystals of pillar[5]arene. We clearly mentioned the point in the introduction part in the revised manuscript.

Comment 2: TITLE: ‘ high mol-weights ‘ . actually, they cannot be considered high molecular weights, but oligomers. Please, erase the concept.

Answer 2: According to the reviewer’s comment, we changed the title as follows:

High-mass fractionation by confinement of polymer in one-dimensional pillar[5]arene channels

Comment 3: C13 NMR is the most powerful tool to detect conformations, dynamics and proof of inclusion. Therefore, it should be enforced considerably with additional experiments. In fact, the demonstration and discussion about conformations in channels is not satisfactory. The authors say: ‘ Both n-alkanes and PEO are linear, unbranched molecules. Thus, these data indicate that PEO prefer to adopt all-anti-like conformations in the PS channel.’ THIS deductive procedure is rather debatable and the discussion too hermetic. Instead, PEO chemical shifts inside the channels must be compared to PEO in the crystalline phase and amorphous phase (which differ considerably). The discussion should be conducted independently of n-hydrocarbons.

Comment 4: The authors state (line : 136): ‘An up-field shift of the carbon signal from PEO was observed for the PSDPEO (Fig. 2b, middle) and P6DPEO complex crystals (Fig. 2b, bottom), which was caused by aromatic shielding of PEO upon incorporation into the complex parts in the crystals formed an assembled structure that was the same as that observed in the X-ray crystal structure of the P6I) n-decane complex; i.e., 1D channels (Fig. 2d). Likewise, P6 molecules also formed a 1D channel containing PEO.’ The analogy with alkanes is not stringent enough. PEO crystallizes in a gauche-containing conformation and its chemical shift in the crystalline state is up-field and not down-field. It must be disentangle clearly the effects of surrounding walls from the conformational arrangement of the macromolecular chains. Instead, n-alkanes crystallize in the all-trans conformation. Additionally, PEO is not fully crystalline, but is necessarily partly amorphous. Which ones are taken as chemical shift for the stretched and the gauche-containing conformations. The MAS spectra, run with CP and without CP at various delay times between pulses, should be performed on the included and confined PEO. Thus, the discussion of the literature and the papers dealing with the 13C NMR chemical shifts and the PEO confinement is to be revised and re-written, by providing more support to the conclusions. The comparison between alkanes of different chain-lengths is confusing and not relevant to the subject of the paper: therefore, the comments on the conformational arrangements of hydrocarbons should be erased.

Answers 3 and 4: According to the reviewer’s comment, we erased the discussion the conformation of PEO chain in 1D channels of **P5** using the X-ray crystal structure between n-alkane and pillar[5]arene. Instead, as a proof of the inclusion, we added magic angle spinning 2D heterocorrelated NMR study in the revision. In **P5** ⊃ PEO complex crystals, the cross peak was observed between the carbon signal of PEO chain and the proton signal of CH₃ group of **P5**, indicating the inclusion of PEO chain in **P5** cavity. We showed the 2D NMR in the revision instead of the X-ray crystal structure. According to the reviewer’s comment, we also added the discussion of the chemical shifts of PEO. From the previous reports, carbon signals of PEO in non-crystalline and crystalline phases were observed at 72.0 and 71.2 ppm, respectively. The chemical shift of neat PEO was observed at 72.0 ppm, indicating formation of crystalline phase. In contrast, the chemical shift observed in **P5** ⊃ PEO complex crystals was ca. 70 ppm, which was remarkably lower than 71.2 ppm and 72.0 ppm assigned to PEO

carbon in non-crystalline and crystalline phases, respectively. Therefore, the upper chemical shift of the **P5**⊃PEO complex crystals would result from aromatic shielding of PEO upon incorporation into the cavities of **P5**. In the revised manuscript, we discussed the points as follows:

The complexation of PEO with **P5** in crystalline state was confirmed directly by magic angle spinning 2D heterocorrelated NMR study (Fig. 2b). The cross peak observed can be assigned to inter-molecular host-guest correlations of the CH₃ proton groups of **P5** and the carbon atoms of PEO, indicating the inclusion of PEO chain in the cavity of **P5**. From the previous reports,^{25,26} carbon signals of PEO in non-crystalline and crystalline phases were observed 72.0 and 71.2 ppm, respectively. The chemical shift of neat PEO was observed at 72.0 ppm (Fig. 2c, top), indicating formation of crystalline phase. In contrast, the chemical shift observed in **P5**⊃PEO complex crystals (Fig. 2c, bottom) was ca. 70 ppm, which was remarkably lower than 71.2 ppm and 72.0 ppm assigned to PEO carbon in non-crystalline and crystalline phases, respectively. Therefore, the upper chemical shift of the **P5**⊃PEO complex crystals would result from aromatic shielding of PEO upon incorporation into the cavities of **P5**.

Comment 5: Molecular weights should be detectible by the analysis of the terminal groups. Solid state NMR should provide evidences between in and out PEO. It is not clear whether conformations are affected by the molecular size and functional end groups.

Answer 5: According to the reviewer comment, we measured molecular weights of PEO. After immersing in the equal-weight mixture of PEO, the integration ratios (main chain of PEO / end group of PEO) were increased, indicating uptake of high mass PEO fraction from the mixture. We showed ¹H NMR spectra and discussed the point the revised manuscript as follows:

The high mass fraction uptake was also confirmed by ¹H NMR (Fig. S12).

About the evidence for the inclusion complex, we showed magic angle spinning 2D hetero-correlated NMR study. About the conformation of PEO, as mentioned above (Answers 3 and 4), discussion of the PEO conformation using solid ¹³C NMR was quite difficult because of the aromatic shielding of PEO and broadening of the PEO signal. We would like to clarify the conformation of PEO using other measurements. This is our next research target. Thank the reviewer to point out the interest.

Comment 6: Line 252: 'half-width at half-maximum (HWHM) values of the peaks for PEO were around 100 Hz and nearly constant at different temperatures (blue squares). The HWHM values for neat PEO (green triangles) and PEO encapsulated in the 1D channel of P6 were almost the same. In stark contrast, the HWHM values for PEO encapsulated in the 1D channel of PS were ca. 300 Hz, and remained almost constant at different temperatures (red circles). This indicates that the mobility of a PEO chain in the 1D channel of PS was much lower than that of a PEO chain in the 1D channel of P6 and in the neat environment.'
On the only basis of the line width is hard to establish the motional behavior: in other words interpreted as diverse T2 (being the reciprocal of the LW). This is a very weak evidence, because the line-width may depend on a few other phenomena differing from mobility. Mobility is indicated as reduced in the channels. On the contrary, increased mobility of

confined macromolecules occurs frequently in compatible channels. The literature on this topic is not discussed.

Answer 6: According to the reviewer's comment, we measured variable temperature T_1 . For the T_1 measurements of the **P5**⊃PEO complex crystals, we used CP-MAS method. However, we could not use CP-MAS method for the T_1 measurements of neat PEO. Therefore, we performed DD-MAS method for the T_1 measurements of neat PEO, which indicated that mobility of neat PEO was remarkably higher than PEO in the 1D channels of **P5**. T_1 values of neat PEO were shorter than that of the **P5**⊃PEO complex crystals, indicating that neat PEO has higher mobility than PEO in the 1D channels of **P5**. The data also suggests that the mobility of a PEO chain in the 1D channel of **P5** was much lower than that of a PEO chain in the neat environment. We mentioned the point in the revised manuscript as follows:

We also measured variable temperature T_1 to investigate the mobility of PEO. For the T_1 measurements of the **P5**⊃PEO complex crystals, we used CP-MAS method. However, we could not use CP-MAS method for the T_1 measurements of neat PEO. Therefore, we performed DD-MAS method for the T_1 measurements of neat PEO, which indicated that mobility of neat PEO was remarkably higher than PEO in the 1D channels of **P5**. In the measurement temperatures, T_1 values of neat PEO were shorter than that of the **P5**⊃PEO complex crystals, indicating that neat PEO has higher mobility than PEO in the 1D channels of **P5**. The data also suggests that the mobility of a PEO chain in the 1D channel of **P5** was much lower than that of a PEO chain in the neat environment.

Comment 7: Additionally, evidence for the formation of IC in P6 with larger channels is weak.

Answer 7: **[redacted]**

Changes made in the manuscript to Reviewer's comments are highlighted in red for clarity (file names: polymer_revision_mark and SI_revison_mark), PDF.

Finally, we wish to thank you and the reviewers for kind and worth comments. We strongly hope that this revised manuscript will be accepted for publication in your journal.

REVIEWERS' COMMENTS:

Reviewer #1 (Remarks to the Author):

I'm satisfied with the changes made and I support the authors' decision to remove the discussion relating to confinement in P6.

A few comments:

I still think that the PXRD pattern of P5/PEO looks quite different from the P5/octane

Scale labels are too small on NMR spectra in Fig2b, S3, and S12.

Reviewer #2 (Remarks to the Author):

The revised version of the paper introduces a few changes while a part of the discussion has been removed. Mistakes and typos have been corrected.

From the experimental point of view, the most relevant improvement was the 2D NMR spectrum, which highlights host-guest interactions. Therefore, the paper is now suitable for publication in Nature Commun., provided the citation nr. 6 and 7 are corrected in the final version. In fact the authors' names are reported incorrectly: the family names are indicated only by the initials, while the first-names appear in the extended form. Please, check accurately also the other references. An akin citation about SINGLE polymer chains in aromatic nano-channels is to be added: Chem. Commun., 7, 2004, 768.

The title, as also observed by ref.1, stresses the concept of 'high molecular weight'. It is still to be changed in something similar to this expression: 'Molecular mass selection by inclusion in molecular crystals....'; without 'high'.

According to two reviewer's comments, we have revised our manuscript. We would like to explain these corrections point-by-point toward each reviewer as follows:

Reviewer 1

Comment 1: I still think that the PXRD pattern of P5/PEO looks quite different from the P5/octane.

Answer 1: The small angle region reflect on the molecular arrangement at nanometer to several nanometer scale, thus we would like to discuss in the small angle region. In the small angle region, the pattern of the P5/PEO looks similar to the P5/octane. We mentioned the point in the revised manuscript as follows:

The PXRD pattern of **P5** \supset PEO complex **in low angles** was similar to that of **P5** \supset *n*-octane complex, but not similar to that of **P5** \supset *n*-hexane complex.

Comment 2: Scale labels are too small on NMR spectra in Fig2b, S3, and S12.

Answer 2: Scale labels in Fig2b, S3, and S12 are enlarged in the revision.

Reviewer 2

Comment 1: the paper is now suitable for publication in Nature Commun., provided the citation nr. 6 and 7 are corrected in the final version. In fact the authors' names are reported incorrectly: the family names are indicated only by the initials, while the first-names appear in the extended form. Please, check accurately also the other references. An akin citation about SINGLE polymer chains in aromatic nano-channels is to be added: Chem. Commun., 7, 2004, 768.

Answer 1: We checked all references and added the Chem Commun paper in the references.

Comment 2: The title, as also observed by ref.1, stresses the concept of 'high molecular weight'. It is still to be changed in something similar to this expression: 'Molecular mass selection by inclusion in molecular crystals....'; without 'high'.

Answer 2: According to the reviewer comment, we corrected the title as follows.

Molecular weight fractionation by confinement of polymer in one-dimensional pillar[5]arene channels

We also corrected the text.